# Seasonality and weather dependance of *Acinetobacter baumannii* complex bloodstream infections in different climates in Brazil

**Sebastião Pires Ferreira Filho[1], Milca Severino Pereira[2], Jorge Luiz Nobre Rodrigues[3], Raul Borges Guimarães[4], Antônio Ribeiro da Cunha[5], José Eduardo Corrente[6], Antônio Carlos Campos Pignatari[7], Carlos Magno Castelo Branco Fortaleza[1]***

1 Department of Infectious Diseases, Botucatu School of Medicine, São Paulo State University (UNESP), City of Botucatu, São Paulo State, Brazil, 2 Department of Nursing, Pontifical Catholic University of Goiás (PUCG), City of Goiânia, Goiás State, Brazil, 3 Center of Health Sciences, Federal University of Ceará (UFC), City of Fortaleza, Ceará State, Brazil, 4 Faculty of Sciences and Technology, Department of Geography, São Paulo State University (UNESP), City of Presidente Prudente, São Paulo State, Brazil, 5 Faculty of Agronomical Sciences, Department of Soil and Environmental Resources, São Paulo State University (UNESP), City of Botucatu, São Paulo State, Brazil, 6 Department of Biostatistics, Botucatu Institute of Biosciences, São Paulo State University (UNESP), City of Botucatu, São Paulo State, Brazil, 7 Department of Medicine, São Paulo Federal University (UNIFESP), City of São Paulo, São Paulo State, Brazil

* carlos.fortaleza@unesp.br

**Data Availability Statement:** All relevant data are within the manuscript and its Supporting Information files.

## Abstract

Recent studies report seasonality in healthcare-associated infections, especially those caused by *Acinetobacter baumannii* complex. We conducted an ecologic study aimed at analyzing the impact of seasons, weather parameters and climate control on the incidence and carbapenem-resistance in *A. baumannii* complex bloodstream infections (ABBSI) in hospitals from regions with different climates in Brazil. We studied monthly incidence rates (years 2006–2015) of ABBSI from hospitals in cities from different macro-regions in Brazil: Fortaleza (Ceará State, Northeast region), Goiânia (Goiás State, Middle-west) and Botucatu (São Paulo State, Southeast). Box-Jenkins models were fitted to assess seasonality, and the impact of weather parameters was analyzed in Poisson Regression models. Separate analyses were performed for carbapenem-resistant versus carbapenem-susceptible isolates, as well as for infections occurring in climate-controlled intensive care units (ICUs) versus non-climate-controlled wards. Seasonality was identified for ABBSI ICUs in the Hospitals from Botucatu and Goiânia. In the Botucatu hospital, where there was overall seasonality for both resistance groups, as well as for wards without climate control. In that hospital, the overall incidence was associated with higher temperature (incidence rate ratio for each Celsius degree, 1.05; 95% Confidence Interval, 1.01–1.09; $P = 0.006$). Weather parameters were not associated with ABBSI in the hospitals from Goiânia and Fortaleza. In conclusion, seasonality was found in the hospitals with higher ABBSI incidence and located in regions with greater thermal amplitude. Strict temperature control may be a tool for prevention of *A. baumanii* infections in healthcare settings.

**Funding:** The authors received no specific funding for this work.

**Competing interests:** The authors have declared that no competing interests exist.

## Introduction

Despite their specific characteristics, healthcare-associated infections (HCAIs) can share some epidemiological determinants with those that occur in the community. Seasonality, increasingly identified in recent studies, is one example [1]. It is prominent in bloodstream infections caused by Gram-negative bacilli (GNB), which have been linked to proximity to the equator [2], summer season [3] and high environmental temperatures measured either within [4] or outside hospitals [5].

This latter aspect is one of the gaps in our current understanding of HCAIs seasonality, since GNB incidence increases during warm periods even within units that are climate-controlled (and thus expected not to present relevant temperature variations) [1,5–7]. Some authors have theorized an influx from reservoirs outside healthcare settings, on the basis of greater seasonality of multidrug-susceptible (as opposed to multidrug resistant, supposedly "hospital-borne") GNB [8] or on molecular heterogeneity of summer strains (which suggests multiple sources) [9]. However, those findings were not supported by other studies, especially those pointing to relevant "summer peaks" of multidrug-resistant GNB infections [6,7,10].

*Acinetobacter baumannii* complex bloodstream infections (ABBSI) stand out among GNB HCAIs for their striking seasonal pattern [11]. They pose therefore unique opportunities to assess HCAIs seasonality. With that in mind, we conducted an ecological study aimed at analyzing how that seasonality of ABBSI varies in different climates, as well as its association with antimicrobial resistance and climate control in hospital units.

## Methods

### Ethical statement

This study adheres to the Helsinki declaration guidelines and was approved by Committee for Ethic in Research (CAAE # 81985517.0.1001.5411) from Botucatu School of Medicine. Since individual data were only used for building time series and most analyses were performed for aggregate data, there was waiver of the application of informed consent forms.

### Study settings

This study was conducted in three hospitals from cities located in different macro-regions of Brazil: Fortaleza (Ceará State, Northeast region), Goiânia (Goiás State, Middle-west) and Botucatu (São Paulo State, Southeast). In all hospitals, Intensive Care Units (ICUs) were climate-controlled, while other wards were not climatized. **Fig 1** presents the geographic distribution of those cities within the map of Brazil. Geographic and climate aspects of those cities are presented in **Table 1**. The study cities differed from each other in the Köppen-Geiger climate classification, according to the Brazil's "National Institute for Spatial Research" (INPE, www.inpe. br) [12].

### Study data

We searched the study hospitals laboratory blood culture databases from years 2006 through 2015. The data were validated in comparison to the hospitals' infection control committees. Briefly, ABBSI were defined as *A. baumannii* recovered from blood cultures collected after 48 hours of patients' admission to the hospitals. Duplicate data (two or more blood cultures from the same patient within a 30-day period) were discarded. After that, time series of monthly incidence (per 10,000 patient-days) were generated for each hospital. Separate rates were calculated for carbapenem-susceptible and carbapenem-resistant *A. baumannii*, as well as for units with or without climate control. Weather parameters (monthly average temperature and

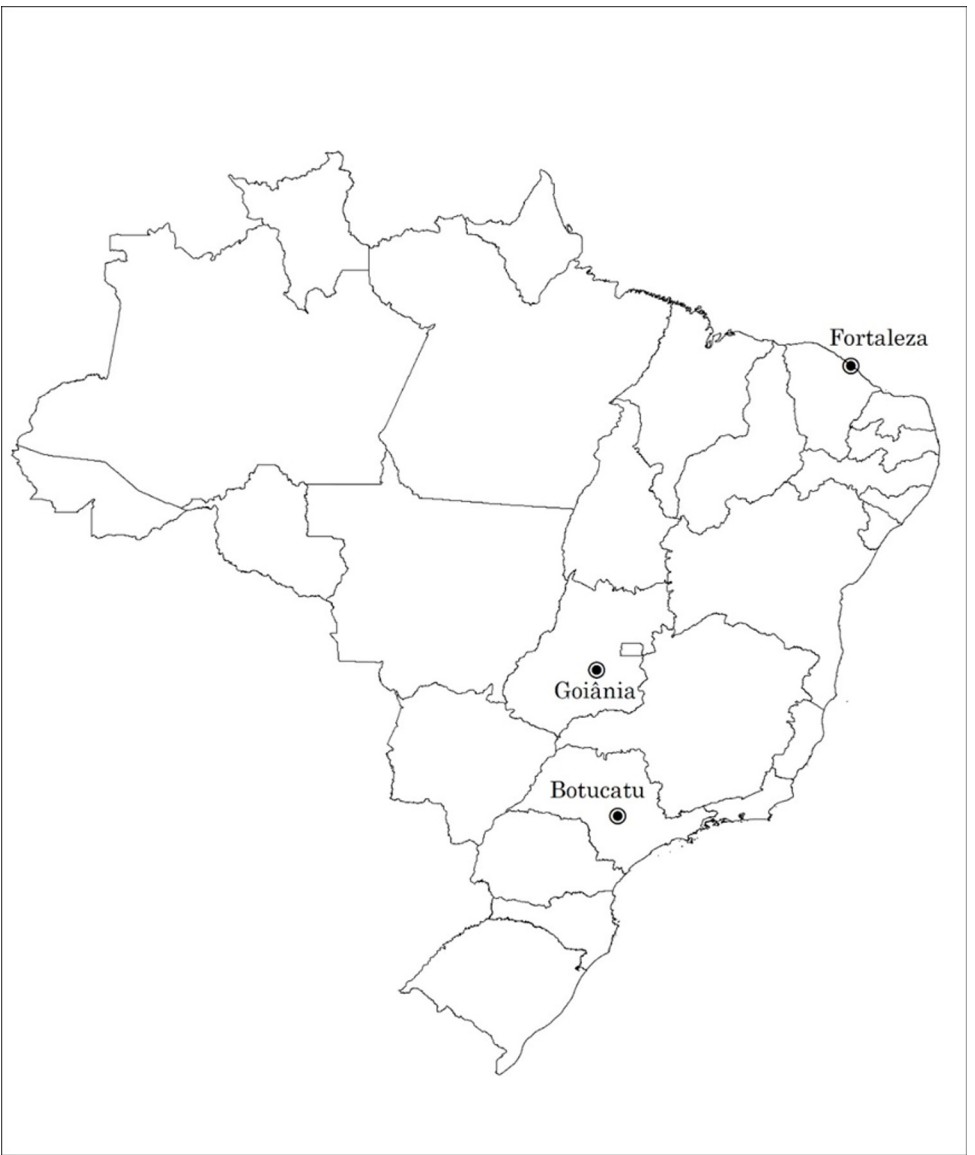

**Fig 1. Map of Brazil, showing the cities that host the study hospitals.** Note. This figure copyrights belong to the authors. Figure was drawn in ArCGis 10 (ESRI, Redlands, CA).

humidity, monthly aggregated rainfall) were obtained from INPE climate database. Of note, raw data for our study are included as S1 File.

## Time series analysis

Incidence rate series were fitted to stochastic, Box-Jenkins models (Seasonal Autoregressive Integrated Moving Average [SARIMA]). Briefly, those models fit time series data for time trends, autoregression (correlation of each value with that recorded in preceding time units) and seasonality (correlation of each value with that recorded in the same period/season from the preceding years) [13]. Therefore, those models allowed us to assess statistically the correlations both with data from immediately preceding months and seasons in preceding years. Models were configured in the usual SARIMA(p,d,q)(P,D,Q)m format using the following

**Table 1. Climatic and geographic characteristics of the municipalities where the study hospitals are located.**

| Hospital | Botucatu Medical School Hospital | Goiás Federal University Hospital | Ceará Federal University Hospital |
|---|---|---|---|
| **City** | Botucatu | Goiânia | Fortaleza |
| **State** | São Paulo | Goiás | Ceará |
| **Macro-region** | Southeast | Middle-west | Northeast |
| **Latitude, longitude** | 22°53′25″S, 48°27′19″W | 16°40′48″S, 49°15′18″W | 3°43′6″S, 38°32′36″W |
| **Altitude (meters above sea level)** | 828 | 764 | 14 |
| **Geographic distance to the Atlantic Ocean (in Kilometers)** | 246 | 868 | 0 |
| **Köppen-Geiger climate classification**[*] | *Cfa* | *Aw* | *As* |
| **Mean monthly temperature in Celsius degrees, average (SE)** | 22.3±3.4 | 23.5±1.5 | 27.4±0.7 |
| **Mean monthly relative humidity, average (SE)** | 72.7±8.7 | 65.8±12.5 | 71.9±6.1 |
| **Monthly total rainfall in millimeters, average (SE)** | 119.4±100.6 | 120.4±110.7 | 103.2±124.9 |
| **Biomes**[**] | Atlantic forest | "Cerrado" | Mangrove/Sandbank |

[*]According to Köppen-Geiger classification[11]: *Cfa*, subtropical climate, humid; *Aw*, tropical climate with dry winter; *As*, tropical with dry summer.

[**]According to the classification of Brazilian Institute for Geography and Statistics (IBGE, www.ibge.gov.br).

trend elements: trend autoregression order (p), 1, i.e., testing the fitness of time series for correlation of each month value with the value recorded one month before; trend difference order (d), 0; trend moving average order (q), 0; seasonal autoregressive order (P), 1 i.e., testing the fitness of time series for correlation of each month value with the value recorded in the same month one year before; seasonal difference order (D), 0; seasonal moving average order (Q), 0; number of time steps (m), 12 (seasonality investigated on a monthly basis). Therefore, we attempted to fit our times series to the models parameters: SARIMA (1,0,0)(1,0,0)12.

## Models of weather dependence

Multivariable Poisson regression models (which are particularly appropriate for outcomes presented as incidence rates) were analyzed with average monthly temperature and humidity, as well as aggregate monthly rainfall as independent variables and monthly rates as outcomes. Both Poisson regression an time series analyses were conducted using STATA 14 software (StataCorp, College Station, TX).

## Results

The aggregate incidence of ABSSI in the study period (for each type of hospital unit and pattern of carbapenem-resistance) is presented in **Table 2,** and time series of monthly incidence is presented in **Fig 2**. Briefly, the Botucatu hospital presented higher overall incidence, but also a greatest proportion of carbapenem-susceptible isolates (especially in years before 2011), when compared to the other two hospitals.

Results from the Box-Jenkins models are presented in **Table 3**. Briefly seasonality was identified for ABSSI in ICUs in the Hospitals from Botucatu and Goiânia. In the Botucatu hospital, there was also overall seasonality for both resistance groups, as well as for wards without climate control.

In the Poisson regression analysis (**Table 4**), we found significant association of temperature with carbapenem-resistant and overall ABBSI in non-climate-controlled units in the

**Table 2. Characteristics of hospitals included in this study, alongside with aggregate incidence of *Acinetobacter baumanii* bloodstream infections (per 10,000 patient-days).**

| Hospital | Botucatu Medical School Hospital | Goiás Federal University Hospital | Ceará Federal University Hospital |
|---|---|---|---|
| **Teaching hospital** | Yes | Yes | Yes |
| **Beds in climate-controlled units** | 98 | 86 | 21 |
| **Beds in non-climate-controlled units** | 462 | 634 | 165 |
| **ABBSI, climate-controlled units\*** | | | |
| *Carbapenem-susceptible* | 10.20 | 1.51 | 0.56 |
| *Carbepenem-resistant* | 12.96 | 16.60 | 6.88 |
| **ABBSI, non-climate-controlled units\*** | | | |
| *Carbapenem-susceptible* | 1.27 | 1.83 | 0.34 |
| *Carbepenem-resistant* | 1.30 | 3.00 | 0.87 |

\*In all study hospitals, Intensive Care Units (ICU) were climate-controlled, while other others were not. ABBSI, *Acinetobacter baumannii* complex bloodstream infections.

Botucatu hospital. There was also a similar association when we used overall hospital incidence of ABBSI as outcome. No association with weather was found for the other study hospitals.

## Discussion

Brazil is a huge country with an area of 8.5 million square kilometers and 200 million inhabitants, distributed 27 states located in five macro-regions. Climates range from equatorial in the Amazon basin to temperate in the South region. There are great socio-economic differences, with poorer areas in the North/Northeast and more developed states in the South/Southeast [14]. Some community-associated infectious diseases are restricted to a climate or biome (e.g., malaria in Amazon), while other occur seasonally in most areas (e.g., Dengue fever) [15]. Among all this complexity, one may wonder how geography and climate impact on the epidemiology of HCAIs occurring in circa 6,000 hospitals in Brazil.

*A. baumannii* complex infections are hyperendemic in Brazilian hospitals [16]. As in other countries, they preferentially affect critically ill patients, undergoing invasive procedures or carrying invasive devices [11,17]. Our study aimed, therefore, at comparing the impact of season and weather on ABBSI incidence in areas that were both geographically distant and presented different climates.

Counterintuitively, ABBSI incidence was higher in the hospital located in the area presenting lower temperatures [2]. Two aspects highlight differences in the epidemiology of ABBSI in the Botucatu hospital. First, continuous incidence is detected, while in other hospitals there were gaps during which *A. baumannii* was not recovered from cultures. (Second, as already mentioned, there was hyperendemicity of carbapenem-susceptible isolates up to year 2010, while in other hospitals more than 90% of isolates recovered all through the study period were carbapenem-resistant.

Seasonality results did not exactly match findings from Poisson regression analysis, using weather parameters as predictors. Some aspects may account for those findings. First, one must notice that weather data were obtained from meteorological stations, located in the same city of (but not inside or nearby) the study hospitals. Therefore, they do not necessarily reflect, for instance, temperature and humidity inside hospital wards. This difference is obviously more relevant for the climate-controlled ICUs, though previous studies from our group have found association of temperature outside hospitals with ABBSI incidence in those units [5,6].

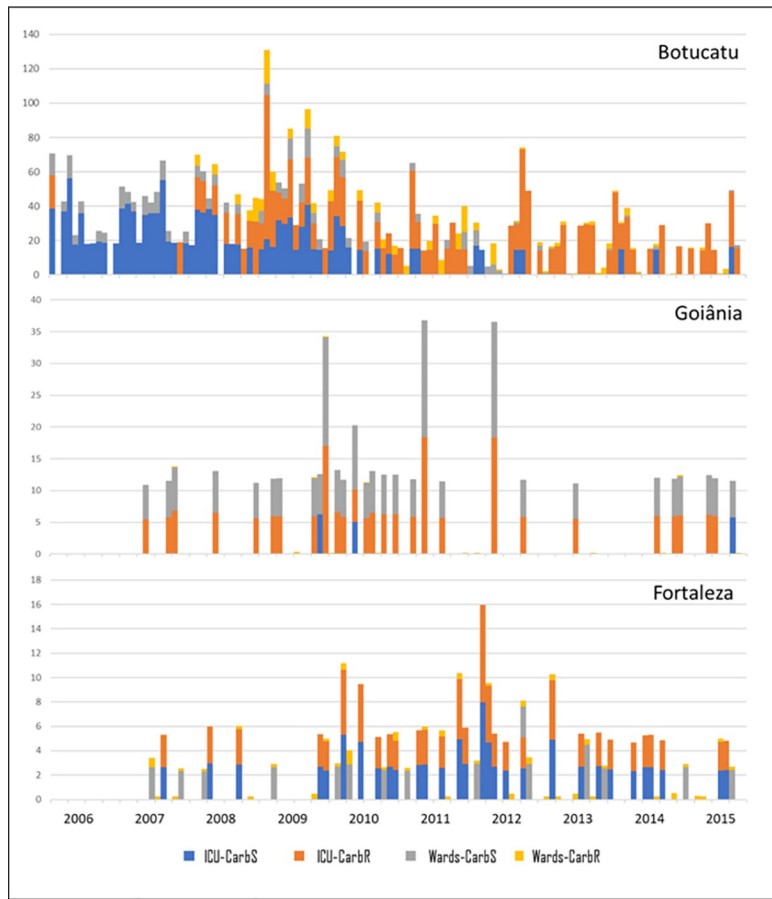

**Fig 2. Monthly incidence of *Acinetobacter baumannii* bloodstream infections in three hospitals from different places in Brazil. Note**. ICU, intensive care units; Wards, units for non-critically ill patients; Carb-R, carbapenem-resistant; CARB-S, carbapenem-susceptible.

Another confounding aspect is the overall difference in the incidence of ABBSI among study hospitals. As one may infer from the Box-Jenkins models presented in **Table 3**, lower incidences hindered subgroup analyses and may have impacted on statistical power of

**Table 3. Seasonal autoregressive coefficients from Box-Jenkins models for incidence of *Acinetobacter baumannii* complex bloodstream infections, according to type of hospital unit and resistance.**

| Hospital Unit, carbapenem susceptibility | Autoregressive coefficient (95% Confidence Interval) | | |
|---|---|---|---|
| | **Botucatu** | **Goiânia** | **Fortaleza** |
| ICU, carbapenem-susceptible | **+0.24 (+0.06 to +0.42)** | -0.03 (-11.34 to +11.29) | +0.09 (-0.05 to +0.23) |
| ICU, carbapenem-resistant | +0.19 (-0.01 to +0.39) | **+0.32 (+0.21 to +0.43)** | +0.09 (-0.05 to +0.22) |
| ICU, total | **+0.25 (+0.09 to +0.41)** | **+0.34 (+0.24 to +0.44)** | +0.09 (- 0.05 to +0.23) |
| Wards, carbapenem-susceptible | -0.01 (-0.16 to +0.17) | + 0.08 (-0.10 to +0.25) | +0.06 (-0.12 to +0.25) |
| Wards, carbapenem-resistant | +0.14 (-0.04 to +0.33) | - 0.06 (-0.42 to + 0.31) | -0.12 (-0.22 to +0.21) |
| Wards, total | **+0.27 (+0.12 to +0.41)** | +0.04 (-0.14 to +0.23) | +0.07 (-0.14 to +0.28) |
| Overall ABBSI | **+0.26 (+0.11 to +0.41)** | + 0.14 (-0.06 to +0.33) | +0.08 (-0.10 to +0.26) |

**Note.** Time series were tested for fitting a Box Jenkins Seasonal Autoregressive Moving Average (SARIMA) model. Statistically significant results ($P < .05$) are presented in boldface. All Intensive Care Units (ICU) were climate controlled, while other wards were not.

**Table 4. Poisson regression analysis of the impact of weather on monthly incidence of *Acinetobacter baumannii* complex bloodstream infections, according to type of hospital unit and resistance.**

| Type of unit/Carbapenem-resistance | Average temperature (ºC) | | Average relative Humidity (%) | | Total rainfall | |
|---|---|---|---|---|---|---|
| | IRR (95%CI) | *P* | RR (IC95CI%) | *P* | IRR (95%CI) | *P* |
| *Botucatu hospital* | | | | | | |
| ICU, carbapenem-susceptible | 1.06 (0.99–1.12) | 0.08 | 0.99 (0.96–1.03) | .96 | 0.99 (0.99–1.01) | 0.22 |
| ICU, carbapenem-resistant | 1.01 (0.95–1.07) | 0.86 | 0.99 (0.96–1.02) | .48 | 0.99 (0.99–1.00) | 0.97 |
| ICU, total ABBSI | 1.03 (0.99–1.08) | 0.19 | 0.99 (0.97–1.01) | .57 | 0.99 (0.99–1.00) | 0.39 |
| Wards, carbapenem-susceptible | 1.03 (0.96–1.11) | 0.43 | 1.00 (0.96–1.04) | .84 | 1.00 (0.99–1.01) | 0.87 |
| Wards, carbapenem-resistant | **1.07 (1.01–1.15)** | **0.04** | 1.00 (0.96–1.04) | .99 | 0.99 (0.99–1.00) | 0.78 |
| Wards, total ABBSI | **1.05 (1.01–1.10)** | **0.04** | 1.00 (0.98–1.03) | .81 | 0.99 (0.99–1.00) | 0.89 |
| Overall ABBSI | **1.05 (1.01–1.09)** | **0.006** | 0.99 (0.98–1.01) | .32 | 0.99 (0.99–1.00) | 0.85 |
| *Goiânia hospital* | | | | | | |
| ICU, carbapenem-susceptible | 2.25 (0.96–5.26) | 0.06 | 1.11 (0.96–1.28) | .15 | 0.98 (0.96–1.03) | 0.09 |
| ICU, carbapenem-resistant | 1.01 (0.79–1.28) | 0.96 | 1.02 (0.98–1.07) | .29 | 0.99 (0.99–1.00) | 0.24 |
| ICU, total ABBSI | 1.08 (0.86–1.36) | 0.48 | 1.03 (0.99–1.07) | .17 | 0.99 (0.98–1.00) | 0.09 |
| Wards, carbapenem-susceptible | 1.23 (0.82–1.87) | 0.31 | 0.96 (0.89–1.02) | .16 | 1.01 (0.99–1.02) | 0.06 |
| Wards, carbapenem-resistant | 0.79 (0.57–1.08) | 0.14 | 0.99 (0.93–1.04) | .61 | 1.00 (0.99–1.01) | 0.33 |
| Wards, total ABBSI | 0.92 (0.73–1.18) | 0.54 | 0.97 (0.93–1.01) | 0.17 | 1.01 (0.99–1.01) | 0.09 |
| Overall ABBSI | 1.00 (0.84–1.17) | 0.97 | 0.99 (0.97–1.03) | 0.97 | 1.00 (0.99–1.00) | 0.89 |
| *Fortaleza hospital* | | | | | | |
| ICU, carbapenem-susceptible | 0.67 (0.11–4.02) | 0.67 | 0.86 (0.62–1.21) | 0.41 | 0.99 (0.97–1.02) | 0.51 |
| ICU, carbapenem-resistant | 1.27 (0.79–20.07) | 0.32 | 0.98 (0.89–1.08) | 0.72 | 0.99 (0.99–1.00) | 0.74 |
| ICU, total ABBSI | 1.21 (0.77–1.93) | 0.41 | 0.98 (0.89–1.06) | 0.57 | 0.99 (0.99–1.00) | 0.67 |
| Wards, carbapenem-susceptible | 1.28 (0.57–2.85) | 0.55 | 0.97 (0.86–1.08) | 0.56 | 1.00 (0.99–1.01) | 0.31 |
| Wards, carbapenem-resistant | 1.17 (0.74–1.88) | 0.49 | 0.99 (0.91–1.08) | 0.88 | 0.99 (0.99–1.01) | 0.81 |
| Wards, total ABBSI | 1.19 (0.80–1.79) | 0.38 | 0.98 (0.91–1.06) | 0.64 | 1.00 (0.99–1.01) | 0.75 |
| Overall ABBSI | 1.20 (0.88–1.63) | 0.24 | 0.98 (0.93–1.03) | 0.45 | 0.99 (0.99–1.00) | 0.99 |

Note. Results with *P*<0.05 are presented in boldface. ABSSI, *Acinetobacter baumannii* complex bloodstream infections; ICU, intensive care units; IRR, incidence rate ratio.

regression models. Also, the variation of temperature among months (and seasons) was greater in the Botucatu (standard error [SE], 3.3) than in Goiânia (SE, 1.5) and Fortaleza (SE, 0.7). That means that the lower the latitude, the lesser the temperature changes though the year.

Finally, there are potential drivers of seasonality which are not directly associated with weather, such summer understaffing [1], the presence of new students and resident doctors or even changes in diseases leading to hospital admission. All those factors require further investigation, though a previous study found no impact of patients' severity-of-illness or comorbidities on the seasonality of Gram-negative (including *A. baumannii*) bloodstream infections [7].

Our study was limited for not individually assessing the average severity-of-illness of patients admitted to different hospitals, as well as their structure for infection control and the quality of microbiology laboratory. However, the three hospitals were included in a recent multistate survey, with similar performance [18]. The same study found minor differences on overall HCAIs incidence among different regions in Brazil [19]. Other limitation was not assessing the incidence of pathogens (e.g., MRSA and GNB) that may compete ecologically for human and inanimate reservoirs inside hospitals [20]. It is therefore possible that the greater incidence of GNB such as *Klebsiella* spp in hospitals close to the Equator line [2] could have impacted on the lower incidence of *A. baumannii* found in our study. Finally, we cannot draw

a representative picture for the whole county with data from only three hospitals. Unfortunately, countrywide surveillance system for nosocomial pathogens was only implemented in 2013, so we did not have access to data from a great number of settings. Rather than providing an overview of *A. baumannii* seasonality in Brazil, we were interested in investigating differences in ABBSI seasonal behavior in hospitals from different areas.

In conclusion, we found seasonality of *A. baumannii* infections in hospitals located in areas with climates ranging from tropical to temperate. Both incidence, seasonality and association with weather were greater in the hospital located in the area with fresher climate and greater temperature range. Those findings reinforce evidence on the seasonal nature of that pathogen. Also, although incidence was greater in climate-controlled ICUs (where patients have severe disease and are exposed to invasive devices) than in non-climate-controlled wards for non-critically ill patients, the association of incidence with higher temperatures was sound. Appropriate climate control is (uncommon in low-to-middle income countries [21]) may be a tool for infection control and prevention.

## Supporting information

**S1 File. Anonymized monthly rates for time series analysis.**
(XLSX)

## Acknowledgments

This study is part of the PhD Thesis of SPFF, with CMCBF as his advisor.

## Author Contributions

**Conceptualization:** Sebastião Pires Ferreira Filho, Antônio Carlos Campos Pignatari, Carlos Magno Castelo Branco Fortaleza.

**Data curation:** Sebastião Pires Ferreira Filho, Milca Severino Pereira, Jorge Luiz Nobre Rodrigues, Raul Borges Guimarães, Antônio Ribeiro da Cunha, José Eduardo Corrente, Carlos Magno Castelo Branco Fortaleza.

**Formal analysis:** Sebastião Pires Ferreira Filho, Jorge Luiz Nobre Rodrigues, Raul Borges Guimarães, Antônio Ribeiro da Cunha, José Eduardo Corrente, Antônio Carlos Campos Pignatari, Carlos Magno Castelo Branco Fortaleza.

**Investigation:** Sebastião Pires Ferreira Filho, Milca Severino Pereira, Jorge Luiz Nobre Rodrigues, Carlos Magno Castelo Branco Fortaleza.

**Methodology:** Sebastião Pires Ferreira Filho, Jorge Luiz Nobre Rodrigues, Carlos Magno Castelo Branco Fortaleza.

**Project administration:** Sebastião Pires Ferreira Filho, Antônio Carlos Campos Pignatari, Carlos Magno Castelo Branco Fortaleza.

**Software:** José Eduardo Corrente.

**Validation:** Sebastião Pires Ferreira Filho, Milca Severino Pereira, Jorge Luiz Nobre Rodrigues, Raul Borges Guimarães, Antônio Ribeiro da Cunha, José Eduardo Corrente, Carlos Magno Castelo Branco Fortaleza.

**Visualization:** Sebastião Pires Ferreira Filho, Milca Severino Pereira, Jorge Luiz Nobre Rodrigues, Raul Borges Guimarães, Antônio Ribeiro da Cunha, José Eduardo Corrente, Carlos Magno Castelo Branco Fortaleza.

**Writing – original draft:** Sebastião Pires Ferreira Filho, Antônio Carlos Campos Pignatari, Carlos Magno Castelo Branco Fortaleza.

**Writing – review & editing:** Sebastião Pires Ferreira Filho, Milca Severino Pereira, Jorge Luiz Nobre Rodrigues, Raul Borges Guimarães, Antônio Ribeiro da Cunha, José Eduardo Corrente, Antônio Carlos Campos Pignatari, Carlos Magno Castelo Branco Fortaleza.

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
