## [Decision Letter · Decision Letter 0]

11 May 2021

PONE-D-21-05493

Seasonality and weather dependance of Acinetobacter baumannii complex bloodstream infections in different climates in Brazil.

PLOS ONE

Dear Dr. Fortaleza,

Thank you for submitting your manuscript to PLOS ONE. After careful consideration, we feel that it has merit but does not fully meet PLOS ONE’s publication criteria as it currently stands. Therefore, we invite you to submit a revised version of the manuscript that addresses the points raised during the review process.

We look forward to receiving your revised manuscript.

Kind regards,

Aleksandra Barac

Academic Editor

PLOS ONE

Journal Requirements:

2. In the ethics statement in the manuscript and in the online submission form, please provide additional information about your retrospective study, including: a) whether you accessed any records containing identifying patient information; b) the date range (month and year) of the study period.

3. Thank you for including your ethics statement: 

"This study adheres to the Helsinki declaration guidelines and was approved by the institutions Committees for Ethics in Research (CAAE # 81985517.0.1001.5411).".   

4. We note that Figure 1 in your submission contain map images which may be copyrighted. All PLOS content is published under the Creative Commons Attribution License (CC BY 4.0), which means that the manuscript, images, and Supporting Information files will be freely available online, and any third party is permitted to access, download, copy, distribute, and use these materials in any way, even commercially, with proper attribution. For these reasons, we cannot publish previously copyrighted maps or satellite images created using proprietary data, such as Google software (Google Maps, Street View, and Earth). For more information, see our copyright guidelines: http://journals.plos.org/plosone/s/licenses-and-copyright.

You may seek permission from the original copyright holder of Figure 1 to publish the content specifically under the CC BY 4.0 license. 

If you are unable to obtain permission from the original copyright holder to publish these figures under the CC BY 4.0 license or if the copyright holder’s requirements are incompatible with the CC BY 4.0 license, please either i) remove the figure or ii) supply a replacement figure that complies with the CC BY 4.0 license. Please check copyright information on all replacement figures and update the figure caption with source information. If applicable, please specify in the figure caption text when a figure is similar but not identical to the original image and is therefore for illustrative purposes only.

Reviewers' comments:

Reviewer's Responses to Questions

**Comments to the Author**

1. Is the manuscript technically sound, and do the data support the conclusions?

Reviewer #1: No

Reviewer #2: Yes

Reviewer #3: Yes

2. Has the statistical analysis been performed appropriately and rigorously? 

Reviewer #1: No

Reviewer #2: Yes

Reviewer #3: Yes

3. Have the authors made all data underlying the findings in their manuscript fully available?

Reviewer #1: No

Reviewer #2: Yes

Reviewer #3: Yes

4. Is the manuscript presented in an intelligible fashion and written in standard English?

Reviewer #1: No

Reviewer #2: Yes

Reviewer #3: Yes

5. Review Comments to the Author

Reviewer #1: Many hospital-acquired blood stream infections worldwide have been attributed to Acinetobacter

baumannii complex. The focus of this study was to assess the impact of seasons, weather parameters

and climate control on the incidence as well as carbapenem-resistance in A. baumannii complex

bloodstream infections in hospitals from three regions of Brazil with different climatic conditions. A

major strength of this manuscript is the use of Box-Jenkins models, which allows for identifying

seasonality in the dependent series (seasonally differencing it if necessary) from even retrospective

datasets. Notwithstanding, the manuscript has a number of flaws that makes it not suitable for

publication in the present state, but of major concern is the methodology section. The data sources

from which the entire results are based on are missing. The author’s stated that monthly incidence rates

between 2006 and 2015 were studied, however failed to show yearly distribution of ABBSI in hospitals

from the study sites. Much emphasis was laid on these raw data but none was showed, not in terms of

charts or tables. Other minor issues have been highlighted in the manuscript.

Reviewer #2: PONE-D-21-05493

Report

In this manuscript the authors analyzed the seasonal changes of ABBSI in different climates, and its relationship with antibiotic resistance and climate control in hospitals. The manuscript seems scientifically sound, and contains some interesting results that can be considered for publication in PONE. However, before the decision of acceptance for publication is running, a very major revison of the manuscript is required. Specifically, the following points should be addressed by the authors:

1、In general, there is a lack of explanation of replicates and statistical methods used in the study. It is expected that the author can give a more detailed explanation or basis why SARIMA is selected in the time series analysis and multivariate Poisson regression models, is selected in the weather-dependent model.

2、Another major question needs to be explained by the author: why only three hospitals are selected, and is the argument for the conclusion sufficient? In addition, whether it is possible to increase the number of hospital cases selected in the same city (that is, under the same climatic conditions) to see if more interesting results can be obtained, and further confirm the conclusion.

Reviewer #3: Review on “Seasonality and weather dependence of Acinetobacter baumannii complex

bloodstream infections in different climates in Brazil.”

The study aims to determine the seasonal weather control of Acinetobacter baumannii complex bloodstream infections (ABBSI) in hospitals from different climatic regions of Brazil. The study conducted hospitals in three climatic zones of Brazil viz. Cfa, subtropical climate, humid (Botucatu São Paulo State, Southeast); Aw, tropical climate with dry winter (Goiânia Goiás State, Middle-west); and As, tropical with dry summer (Fortaleza, Ceará State, Northeast region). The values of monthly averaged climate parameters viz. temperature, relative humidity, and rainfall have been obtained from the INPE climate dataset. The monthly incidence rates of ABBSI (per 10,000 patient-days) have been calculated for Carbapenem-susceptible and Carbapenem-resistant cases separately and for climate control units and non-climate control units. The analysis using Seasonal Autoregressive Integrated Moving Average has been made. The study is novel and valuable. It would have a wider readership if it published in PLOS-One. Having said that, I feel the study requires major revision. My comments are mentioned below.

1. It is not clearly stated in the manuscript that why the authors intended to do such a study. The motivation and significance of the study need to be brought out clearly in the manuscript before its publication.

2. The authors have selected the hospitals in different climatic zone to understand the impact of the weather on ABBSI. The elevation and proximity of the ocean differ for all these stations. The author needs to justify how their experiment design does not affect by these factors in the revised manuscript. For the better readership of this paper, it will be helpful if authors supplement a detailed description of the SARIMA model in the revised manuscript.

3. Table 2 shows that ABBSI per (10000 patients-days) are more in climate control units than non-climate control units for Carbapenem-susceptible and Carbapenem-resistant cases. However, in conclusion, they have stated that “Strict temperature control may be a tool for prevention of A. baumanii infections in healthcare settings.” – It is confusing and need proper analysis and explanation.

4. The temperature, relative humidity, and rainfall have been analyzed to understand their impact on ABBSI incidences. Relative humidity and rainfall do not vary significantly over all three stations, as seen from the monthly average dataset. Therefore the variation of temperature has importance. However, it will be valuable if authors perform this study for non-rainy months and rainy months.

The author needs to check spelling and grammar before submitting the revised manuscript. There are some typo-grammatical errors in the manuscript.

6. PLOS authors have the option to publish the peer review history of their article (what does this mean?). If published, this will include your full peer review and any attached files.

Reviewer #1: No

Reviewer #2: No

Reviewer #3: No

---

## [Author Response · Author response to Decision Letter 0]

13 Jul 2021

RESPONSE TO REVIEWERS

To the Editor and Reviewers – 

We thank you for extensively reading our manuscript and for your excellent comments. Whenever possible, we followed your recommendations and changed the manuscript accordingly. Changes in the revised manuscript are presented in red types (in the “track changes version”). We address each comment and recommendation bellow.

Editorial team

 We note that Figure 1 in your submission contain map images which may be copyrighted. All PLOS content is published under the Creative Commons Attribution License (CC BY 4.0), which means that the manuscript, images, and Supporting Information files will be freely available online, and any third party is permitted to access, download, copy, distribute, and use these materials in any way, even commercially, with proper attribution. For these reasons, we cannot publish previously copyrighted maps or satellite images created using proprietary data, such as Google software (Google Maps, Street View, and Earth). For more information, see our copyright guidelines: http://journals.plos.org/plosone/s/licenses-and-copyright.

Authors’ response: Figure 1 was generated in ArcGis (ESRI, Redlands, CA) and its copyrights belong to the authors. We included that information as a note in the figure.

Reviewer #1

Reviewer #1: Many hospital-acquired blood stream infections worldwide have been attributed to Acinetobacter baumannii complex. The focus of this study was to assess the impact of seasons, weather parameters and climate control on the incidence as well as carbapenem-resistance in A. baumannii complex bloodstream infections in hospitals from three regions of Brazil with different climatic conditions. A major strength of this manuscript is the use of Box-Jenkins models, which allows for identifying seasonality in the dependent series (seasonally differencing it if necessary) from even retrospective datasets. Notwithstanding, the manuscript has a number of flaws that makes it not suitable for publication in the present state, but of major concern is the methodology section. The data sources from which the entire results are based on are missing. The author’s stated that monthly incidence rates between 2006 and 2015 were studied, however failed to show yearly distribution of ABBSI in hospitals from the study sites. Much emphasis was laid on these raw data but none was showed, not in terms of charts or tables. Other minor issues have been highlighted in the manuscript.

Authors’ response: We thank Reviewer #1 for her/his comments. The data used for analysis were obtained in laboratory files in the study hospitals, and validated in accordance to databases from the infection control committee’s in every study hospital. We clarified that topic in the methodology section. A new figure (Figure 2) presenting time series of monthly ABBSI incidence was included. Also, we included a statement informing that raw data are submitted as supplementary file. We included the type of study in the abstract (ecologic study).

Reviewer #2

In this manuscript the authors analyzed the seasonal changes of ABBSI in different climates, and its relationship with antibiotic resistance and climate control in hospitals. The manuscript seems scientifically sound, and contains some interesting results that can be considered for publication in PONE. However, before the decision of acceptance for publication is running, a very major revison of the manuscript is required. Specifically, the following points should be addressed by the authors: 1、In general, there is a lack of explanation of replicates and statistical methods used in the study. It is expected that the author can give a more detailed explanation or basis why SARIMA is selected in the time series analysis and multivariate Poisson regression models, is selected in the weather-dependent model. 2、Another major question needs to be explained by the author: why only three hospitals are selected, and is the argument for the conclusion sufficient? In addition, whether it is possible to increase the number of hospital cases selected in the same city (that is, under the same climatic conditions) to see if more interesting results can be obtained, and further confirm the conclusion.

Authors’ response: We expanded explanations on the reasons for using Box-Jenkins models and Poisson regression. We used only three hospitals because, previously to 2013, there was no countrywide data on incidence of healthcare-associated pathogens. Those three hospitals agreed to share their data. We are aware that this does not provide a wide or representative sample of the country, but those were the data available for our analysis. We attempted to compare those hospitals from different areas rather than providing a representative overview of ABBSI incidence in Brazil. We included this limitation in the discussion section.

Reviewer #3

The study aims to determine the seasonal weather control of Acinetobacter baumannii complex bloodstream infections (ABBSI) in hospitals from different climatic regions of Brazil. The study conducted hospitals in three climatic zones of Brazil viz. Cfa, subtropical climate, humid (Botucatu São Paulo State, Southeast); Aw, tropical climate with dry winter (Goiânia Goiás State, Middle-west); and As, tropical with dry summer (Fortaleza, Ceará State, Northeast region). The values of monthly averaged climate parameters viz. temperature, relative humidity, and rainfall have been obtained from the INPE climate dataset. The monthly incidence rates of ABBSI (per 10,000 patient-days) have been calculated for Carbapenem-susceptible and Carbapenem-resistant cases separately and for climate control units and non-climate control units. The analysis using Seasonal Autoregressive Integrated Moving Average has been made. The study is novel and valuable. It would have a wider readership if it published in PLOS-One. Having said that, I feel the study requires major revision. My comments are mentioned below. 1. It is not clearly stated in the manuscript that why the authors intended to do such a study. The motivation and significance of the study need to be brought out clearly in the manuscript before its publication. 2. The authors have selected the hospitals in different climatic zone to understand the impact of the weather on ABBSI. The elevation and proximity of the ocean differ for all these stations. The author needs to justify how their experiment design does not affect by these factors in the revised manuscript. For the better readership of this paper, it will be helpful if authors supplement a detailed description of the SARIMA model in the revised manuscript. 3. Table 2 shows that ABBSI per (10000 patients-days) are more in climate control units than non-climate control units for Carbapenem-susceptible and Carbapenem-resistant cases. However, in conclusion, they have stated that “Strict temperature control may be a tool for prevention of A. baumanii infections in healthcare settings.” – It is confusing and need proper analysis and explanation. 4. The temperature, relative humidity, and rainfall have been analyzed to understand their impact on ABBSI incidences. Relative humidity and rainfall do not vary significantly over all three stations, as seen from the monthly average dataset. Therefore the variation of temperature has importance. However, it will be valuable if authors perform this study for non-rainy months and rainy months.

 The author needs to check spelling and grammar before submitting the revised manuscript. There are some typo-grammatical errors in the manuscript.

Authors’ response: 1. We included the following statement in the discussion: Rather than providing an overview of A. baumannii seasonality in Brazil, we were interested in investigating differences in ABBSI seasonal behavior in hospitals from different areas. We believe that this statement clarifies the intention of our study (as well as its limitations, such as those provided by the study design). 2. We included a more detailed description of the SARIMA model in the methods section (please see response to Reviewer #2). 3. The incidence was higher in climate controlled units because they were intensive care units, therefore harboring more severilly ill patients with invasive devices. However, we changed the conclusion section in order to avoid confusing messages. 4. We did preliminar analyes of rainy vs. non-rainy months and colder vs warmer seasons. They presented no statistical significance and did not relevantly expand our inferences. So we preferred to exclude them. As for typos and other errors, we performed an extensive revision of the manuscript. We thank Reviewer #3 for her/his comments and recommendations.

We believe that the peer review process is a rich opportunity for important discussion on scientific relevance and methodology, and that reviewers’ comments and recommendations helped us improving our manuscript. We thank once again the reviewers and editor.

Yours,

The authors

---

## [Editor Report · Decision Letter 1]

21 Jul 2021

Seasonality and weather dependance of Acinetobacter baumannii complex bloodstream infections in different climates in Brazil.

PONE-D-21-05493R1

Dear Dr. Fortaleza,

We’re pleased to inform you that your manuscript has been judged scientifically suitable for publication and will be formally accepted for publication once it meets all outstanding technical requirements.

Kind regards,

Aleksandra Barac

Academic Editor

PLOS ONE
---

## [Editor Report · Acceptance letter]

10 Aug 2021

PONE-D-21-05493R1 

Seasonality and weather dependance of *Acinetobacter baumannii* complex bloodstream infections in different climates in Brazil. 

Dear Dr. Fortaleza:

I'm pleased to inform you that your manuscript has been deemed suitable for publication in PLOS ONE. Congratulations! Your manuscript is now with our production department. 

Kind regards, 

on behalf of

Dr. Aleksandra Barac 

Academic Editor

PLOS ONE